# Preparation, Microstructure, Mechanical Properties and Biocompatibility of Ta-Coated 3Y-TZP Ceramic Deposited by a Plasma Surface Alloying Technique

**DOI:** 10.3390/ma13061265

**Published:** 2020-03-11

**Authors:** Ke Zheng, Liangliang Li, Yaqian Dong, Jie Gao, Hongjun Hei, Yong Ma, Bin Zhou, Zhiyong He, Yongsheng Wang, Shengwang Yu, Bin Tang, Yucheng Wu

**Affiliations:** Institute of New Carbon Materials, Taiyuan University of Technology, Taiyuan 030024, China; zhengke0719@163.com (K.Z.); liliangliang@163.com (L.L.); dongyaqian0713@163.com (Y.D.); gaojie@tyut.edu.cn (J.G.); heihongjun@tyut.edu.cn (H.H.); mayong@tyut.edu.cn (Y.M.); zhoubing@tyut.edu.cn (B.Z.); hezhiyong@tyut.edu.cn (Z.H.); wangyongsheng@tyut.edu.cn (Y.W.); tangbin@tyut.edu.cn (B.T.); wuyucheng@163.com (Y.W.)

**Keywords:** 3Y-TZP, Ta coating, hydrothermal aging, biaxial flexural strength, biocompatibility

## Abstract

A Ta coating has been successfully fabricated on the surface of zirconia polycrystals ceramic (3 mol% yttria, 3Y-TZP) by a plasma surface alloying technique. The X-ray diffraction (XRD) and scanning electron microscopy (SEM) results showed that a *α*-Ta coating with a continuous and compact surface morphology which consisted of a deposited layer with a thickness of 390 nm and a diffusion layer with a thickness of 200 nm covered the 3Y-TZP. Due to the effect of inhabitation the *t*→*m* transformation by the deposited Ta coating, the biaxial flexural strength caused by the phase transformation during hydrothermal aging is reduced e.g., *p* < 0.05 after 20 h and/or 100 h. In addition, the Ta coating shows non-cytotoxicity and improved proliferation ability of osteoblasts.

## 1. Introduction

The use of zirconia ceramics as implantodontics and bone prostheses [1,2], has received great attention due to their excellent properties. However, zirconia has three phases, including the monoclinic phase (that is *m*), tetragonal phase (*t*) and the cubic phase (*c*) [3,4]. Due to the lack of toughness of the pure *m*-phase, some oxides such as Y_2_O_3_ were added to the pure zirconia for introducing the tetragonal and cubic phase [5,6]. The metastable *t*-phase would undergo a *t*→*m* transformation induced by external loading at room temperature, resulting in an increasing volume which impedes and delays the rapid propagation of main cracks, corresponding to phase transformation toughening [3]. However, some factors including friction, an acid-base environment and load patterns could promote this *t*→*m* transformation at low temperature [7], which would negatively influence the mechanical properties [4] e.g., yielding strength, toughness, etc.

The fabrication of a protective layer to inhibit and postpone the starting phase transformation of *t*→*m*, has attracted a great deal of attention. Because the thin protective layer is inert to the component’s dimensional accuracy, and the substrate’s mechanical characteristics [8], metals and alloys such as Ti [9,10], and CoCr [11] have been employed as coatings on ceramic substrates. Among these materials, tantalum (Ta) coatings have attracted attention for biomedical applications because of their excellent anti-corrosion properties [12], in vivo bioactivity [13,14,15], and plasticity as well as toughness [16], which have been exploited in implant fixtures. However, the pure Ta with high melting point is hard to coat on the surface of ceramics by traditional processes. The plasma surface alloying technique (PSAT) [17], a development from both plasma nitriding and sputtering techniques, has been used to apply many solid elements onto the surface of various components and substrate materials to achieve excellent surface properties. The metal as the target material is bombarded by a glow discharge to initiate the production of active atoms which are deposited on the surface of substrates, forming a diffusion layer at the interface. Because this technique can improve the adhesion between the coating and the substrate, it has been successfully utilized to fabricate surface-alloyed layers or coatings to improve the surface properties (against wear resistance, anti-corrosion and oxidation resistance) of diverse substrate materials, e.g., steel, iron, titanium alloy and intermetallic compounds.

In the current study, a Ta coating was deposited onto zirconia polycrystals containing 3 mol% yttria (3Y-TZP) using the plasma surface alloying technique as a protective layer on 3Y-TZP to suppress the *t*→*m* phase transformation. The microstructure, and morphology were studied. Moreover, the mechanical properties and biocompatibility of the Ta-coated 3Y-TZP specimens were investigated too.

## 2. Materials and Methods

### 2.1. Preparation

A 3Y-TZP ceramic specimen with dimensions of 14 mm in diameter and 1.2 mm in thickness was fabricated [18]. The grains size of 3Y-TZP is ~ 100–300 nm, as shown in Figure 1. The specimens were cleaned in acetone and alcohol for 30 min, respectively, and finally dried using hot air. The coatings were deposited by using the plasma surface alloying technique [17,19]. As the target metal Ta (purity ~ 99.99 wt. %) with dimensions of 60 mm (diameter) × 1.5 mm thickness was deposited on the 3Y-TZP ceramic substrate under a protective high purity argon gas stream (purity ~ 99.999%). The experimental temperature was examined by an infrared thermometer. More detailed experimental parameters are listed in Table 1.

### 2.2. Characterization 

X-ray diffraction (XRD, DX-2700, Dandong Haoyuan Instrument, Dandong, China) with Cu-Ka radiation was utilized to examine the microstructure of the as-received 3Y-TZP ceramic substrate and Ta coating specimen. Surface and cross-section morphologies of as-received 3Y-TZP ceramic substrate and Ta coating specimen were characterized by scanning electron microscopy (SEM, TESCAN MIRA3 LMH, Brno, Czech Republic), and depth-dependent profiles of elements were examined by energy dispersive X-ray spectroscopy (EDS). The Ta coating specimen was selected to prepare the specimen by a focused ion beam (FIB, Nanolab 600i, FEI, Portland, OR, USA) for the ESD chemical composition analysis. The water contact angle was tested by a Contact Angle Surface Analyzer (LSA200, Eastern-Dataphy Instrument, LAUDA Scientific, Lauda-Königshofen, Germany).

### 2.3. Performance Tests 

The protection effect of Ta coatings to inhibit the *t*→*m* phase transformation was studied by successive hydrothermal aging of the specimens in an autoclave (Type LS-30, BoXun, Shanghai, China) for up to 100 h in water vapor (134 °C) and under 3 bar pressure. The progress of the *t*→*m* phase transformation in uncoated and Ta coated specimens was observed at 20 h and 100 h. In order to study the protection effect of Ta coating and different hydrothermal aging time on 3Y-TZP, the biaxial flexure strength was measured according to ISO-6872. The test procedure in detail can be found in [20]. The transformation region of *t*→*m* in specimen after aging 100 h was observed by SEM. Each specimen was tested three positions to ensure the accuracy of the results. The white light interferometer (contour GT-X3, Bruker, Saarbrücken, Germany) was utilized to test the surface roughness of uncoated and Ta coated 3Y-TZP. It had found that the surface roughness of as-received 3Y-TZP was of 161.1 nm, but decreased to 108.3 nm after deposited 800 °C for 10 min [21].

The MG-63 preosteoblastic cells (obtained from the cell bank of the Chinese Academy of Sciences, Shanghai, China) were utilized to evaluate the cytocompatibility of the Ta coating on 3Y-TZP. The density of the cell cultured on each sample is 4 × 10^4^ cells per cm^2^ in the cytotoxicity and proliferation assays. A Live/Dead Viability/Cytotoxicity kit for cells (Invitrogen, Shanghai, China) was utilized to determine the cytotoxicity of the samples. After culturing for 1, 3 and 5 days, the specimens were rinsed thrice with sterilized phosphate buffer solution (PBS) and stained by Live/Dead fluorescent dye for 40 min, and then observed and taken pictures by using a laser scanning confocal microscope (CLSM, C2 Plus, Nikon, Toyota, Japan). A Cell Counting Kit-8 (CCK-8, Beyotime, Shanghai, China) was used to investigate osteoblast proliferation on Ta-coated 3Y-TZP ceramic. After incubation of 1, 3 and 5 days, the specimens were gently rinsed trice with PBS. Then 900 µL culture medium and 100 µL CCK-8 dye were added to form a water soluble adduct. After 4 h, the obtained solutions were transferred to a 96 well plants, and the optical density of each well was tested at 450 nm on a microplate reader (384 Plus, Molecular Devices, San Jose, CA, USA). As for the cell culture and subculture process, readers can refer to [22] for more details.

### 2.4. Statistical Analysis

In the qualitative test, representative picture of each group was choose from six randomly captured fields. In quantitative assays, three samples were used in each group for the stable results, which were displayed as mean ± standard deviation. The Statistical Product and Service Solutions software (SPSS Version 19.0, International Business Machine IBM, New York, NY, USA) was employed to analysis experimental results statistically. The statistical significance between each group was analyzed by one-way ANOVA followed by the Student-Newman-Keuls test. The difference was considered to be significant when *p* < 0.05.

## 3. Results and Discussion

### 3.1. Microstructure, Surface Morphology and Composition 

Figure 2 shows the XRD results of the as-received and Ta-coated 3Y-TZP specimens. For the deposited Ta coating spacimen, strong diffraction peaks at 38.4°, 55.5°, 69.5° and 82.4° appear, which correspond to (110), (200), (211) and (220) of α-Ta phase indicating the body-centered-cubic structure [23]. The broadened peaks of Ta may be related to an amorphous phase in the coating. Because the thickness of Ta interlayer is less than 1 µm, ZrO_2_ diffraction peaks were also detected.

Figure 3 displays the surface and cross-section morphology as well as the corresponding elemental distributions of Ta-coated 3Y-TZP specimens at 800 °C. The Ta coating shows a continuous and compact morphology, which consists of nanocrystalline grains of an average grain size of ~ 60 nm with a pyramid-like morphology, as shown in Figure 3a. An obvious interface between the 3Y-TZP substrate and Ta coating with the tight connecting surface can be observed. Moreover, the Ta/3Y-TZP interface is free of cracks. The thickness of the deposited-Ta coating is of ~ 390 nm (Figure 3b). An elemental diffusion zone exists at the Ta/3Y-TZP interface. Clearly, the Zr, O and the Ta elements diffused toward each other regardless of the element Y. The existence of elementa O in the coating can be attributed to the oxidation of Ta in air [22].

Details of the elemental distribution were investigated by EDS line scan results, as shown in Figure 4. The intensity of Ta distribution in its diffusion zone decreases gradually, while the intensity of O and Zr elements increases (Figure 4a), indicating the excellent adhesion between the Ta coating and the substrate. The diffusion zone width of elemental Ta is of ~ 200 nm, but the elements O and Zr show a narrow width of ~ 100 nm. Figure 4b shows the elemental diffusion of Ta, Zr and O in the diffusion zone, respectively. Here, the non-steady diffusion was caused by bombardment of argon ions during the fabrication process of Ta coating, so Fick’s second law was employed to evaluate the non-equilibrium diffusion between the Ta coating and the substrate [17,24]:(1)∂C∂t=D(∂2C∂x2)
where *D*, *t* and *x* are the diffusion coefficient, time and location, respectively. The concentration of any diffusing location, *C*, could be got as: (2)$$C(x,t)=\frac{2M}{\sqrt{\pi Dt}}\exp[-\frac{{\left(x-x_{0}\right)}^{2}}{4Dt}]$$

Here, the *D* of Ta, Zr and O are of 3.86 × 10^−6^ μm^2^/s, 8.58 × 10^−6^ μm^2^/s and 7.41 × 10^−6^ μm^2^/s. *M* and *x*_0_ are constant. Yoshimura et al. [25] proposed that a vacancy gradient layer with a high concentration of defects was created by the Ar ion bombardment effect at the substrate surface. By the combination with the thermal gradient effect on the substrate, these defects as a diffusion channel combination with the thermal gradient effect on the substrate surface promote the deposited element atoms’ diffusion, leading to the large diffusion coefficient.

Figure 5 shows the water contact angle of uncoated 3Y-TZP substrate and a Ta-coated specimen. The water contact angle was 41° for uncoated 3Y-TZP substrate, but decreased to 28.5° for Ta-coated specimen.

### 3.2. Effect of Ta Coatings

Figure 6 displays the effect of Ta coating on the biaxial flexural strength of 3Y-TZP after hydrothermal aging. The flexural strength of 3Y-TZP substrate depends strongly on the aging duration. Before the hydrothermal aging, the flexural strength of as-recieved specimen was of 1322 MPa. After hydrothermal aging for 20 h, it dropped to 1026 MPa, a 22.4% reduction. Moreover, after hydrothermal aging for 100 h, the biaxial flexural strength decreased to 756 MPa, reducing by 42.8%. One possible reason is that the *m*-phase volume fraction increased through the phase transformation of *t*→*m* increased during hydrothermal aging [26,27]. Our previous reports found that the monoclinic phase content could increase 75% after 100 h hydrothermal aging [21]. In comparison, the flexural strength of Ta coated specimens displayed the weak dependence on the aging duration. The flexural strength dropped from 1350 MPa to 1290 MPa, decreasing by 4.4% after hydrothermal aging for 20 h. Moreover, it lowered to 1045 MPa, dropped by 22.6%. The flexural strength of uncoated and coated specimens at 20 h of aging and at 100 h of aging had significant difference (^★^*p* < 0.05 compared to control). Due to the prohibition phase transformation of *t*→*m* by Ta coatings, the content of monoclinic phase of Ta coated specimens was of 31% after 100 h hydrothermal aging [21,26].

Yoshimura et al. [25] supposed the OH^−^ diffusion in the lattice to be responsible for the *t*→*m* transformation. After water adsorption on the 3Y-TZP surface, the generation of Zr–OH (Y–OH) at the surface of zirconia atoms promote nucleation and growth of the *m*-phase, which proceeds to the inside of the bulk 3Y-TZP [15,27,28]. The Ta coating impedes the direct contact between the water molecules in the environment and the surface of 3Y-TZP, implying that water constituents would find it harder and need to pass a longer distance to diffuse into the near-surface 3Y-TZP, so the protective effect of Ta coating shows notable advantages to improve the mechanical properties.

Figure 7 shows the SEM morphology of the transformation region after 100 h hydrothermal aging. The X-ray penetration depth can be up to 5–10 µm, so it can only be used to detect the near-surface phase transitions of zirconia [21]. The visible interface does not represent the real interface of the *t*→*m* phase transition region, but rather depicts the location with the rising depth of material and reducing content of *m*-phase, intergranular stresses leading to volume gain of transformed 3Y-TZP that can hardly contribute to intergranular fracture [5]. Clearly, the near surface region is characterized by a typical intergranular fracture morphology, full of microcracks, rough, sharp edges and surface grains. In comparison, the transgranular fracture for the untransformed 3Y-TZP results in a flatter surface with no obvious grain interface [5], so the distance between the boundary line of the two regions and the surface can be regarded as the transition region. After hydrothermal aging for 100 h, the transformed distance of uncoated specimens is ~ 45 µm, but in Ta- coated specimens this is extended by ~ 10 µm, respectively. Thus indicates a notable inhabitation of the *t*→*m* phase transformation. 

With the increase of the volume of monoclinic phase, the stress concentration would be caused by the phase transition in the local region around the surface defects, which can promote the formation of pores or voids and the propagation of critical cracks [29]. These microcracks will propagate under the external loading, which may result in the final fracture failure, so the mechanical properties such as the strength and fracture toughness can be improved by impeding the low temperature degradation of 3Y-TZP structures. 

### 3.3. Cytocompatibility of Uncoated and Ta Coated 3Y-TZP

Figure 8 shows the cytotoxicity results of 3Y-TZP specimens with and without Ta coating. After culturing for 1, 3 and 5 days, no obvious dead cells can be seen, which indicates that both 3Y-TZP substrate and Ta-coated 3Y-TZP are non-cytotoxic. After 3 and 5 days, the cell density (fluorescence intensity and density) on Ta-coated 3Y-TZP increases visually compared to the control group. The cytotoxicity results are in accordance with the cell proliferation assay results (Figure 9). After 1 and 3 days, the Ta-coated 3Y-TZP group has higher optical density (OD) values than the control one. After 5 days, the osteoblast proliferation ability on Ta-coated 3Y-TZP group is significantly enhanced, which means the non-cytotoxic Ta coating can help to improve the proliferation ability of 3Y-TZP ceramics.

Cells respond to implants differently according to their surface topography and chemical status [30]. Since there are no cytotoxicity elements in 3Y-TZP ceramic, and thus no apparent dead cells can be seen, this proves that 3Y-TZP is a cell-friendly implant. However, few studies have figured out whether Ta coating is more bioactive than the widely used bulk 3Y-TZP. In this work, Ta-coated specimens and the 3Y-TYP substrate possess a similar nano-topography (Figure 1), and close surface roughness (161.1 nm vs. 108.3 nm), but the hydrophilicity of the substrate is notably improved after coating by Ta. Researchers have found that cell-material interactions can be improved by achieving a high surface energy and excellent wettability [31,32]. A hydrophilic surface will promote cell attachment, adhesion, spreading, proliferation and other activities [33], so we can come to the conclusion that the flatter surface and improved hydrophilicity all contribute to the enhanced proliferation ability of Ta-coated 3Y-TZP.

## 4. Conclusions

In this work, a Ta coating was applied onto the 3Y-TZP surface via PSAT. It is found that the continuous and compact Ta-coating with a body-centered-cubic structure consisted of a deposited and a diffused layer. The thicknesses of the deposited layer and diffused layer are ~ 390 nm and ~ 200 nm, respectively. The biaxial flexural strength of as-received 3Y-TZP specimens has a strong dependence on hydrothermal aging due to the *t*→*m* transformation during aging, whereas, the Ta coating can effectively inhibit the *t*→*m* transformation, and slow down the decrease of flexural strength and the decrease of the flexural strength caused by the phase transformation during hydrothermal aging is thus reduced significantly (*p* < 0.05) after 20 h and/or 100 h. Moreover, the Ta- coated 3Y-TZP specimens shows no cytotoxicity and improved osteoblast proliferation ability.

## Figures and Tables

**Figure 1 materials-13-01265-f001:**
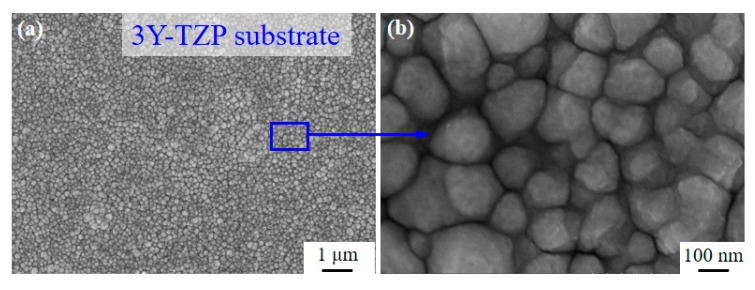
(**a**) SEM images of the 3Y-TZP ceramic, and (**b**) image with large magnification.

**Figure 2 materials-13-01265-f002:**
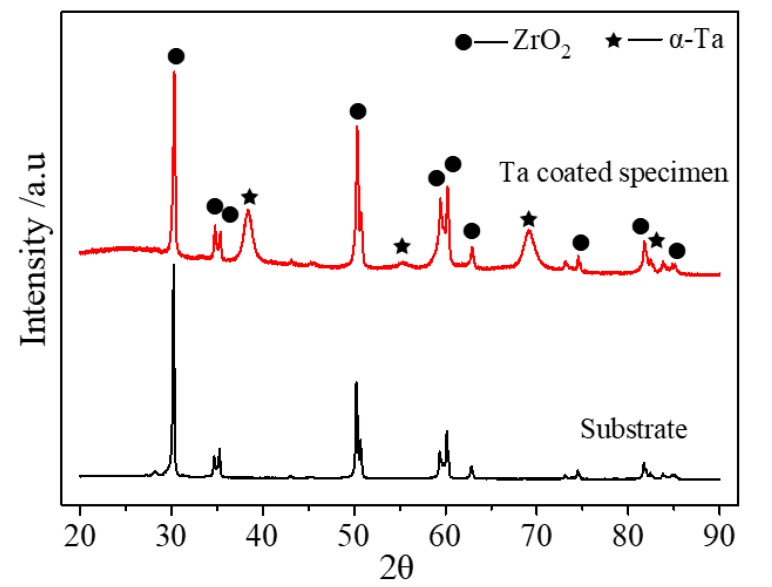
XRD profiles of as-received 3Y-TZP substrate and Ta-coated specimens.

**Figure 3 materials-13-01265-f003:**
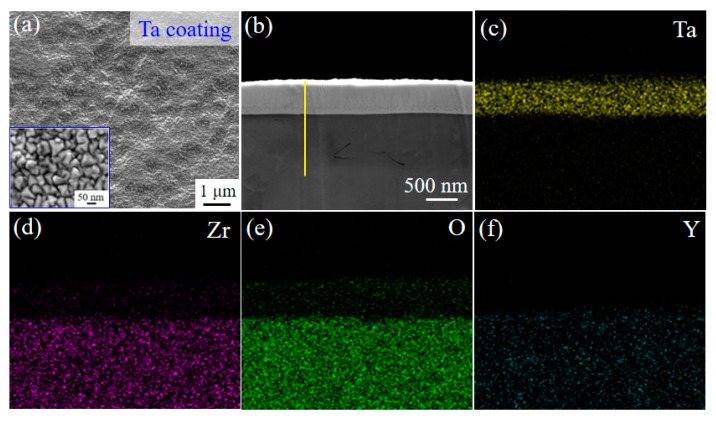
(**a**) Surface morphology of the 3Y-TZP substrate and its enlarged morphology inset. (**b**) The cross-section morphology of the Ta coated specimen, where the yellow line in (**b**) is utilized to line composition analysis in Figure 4; and (**c**–**f**) the corresponding elemental distributions from mapping results.

**Figure 4 materials-13-01265-f004:**
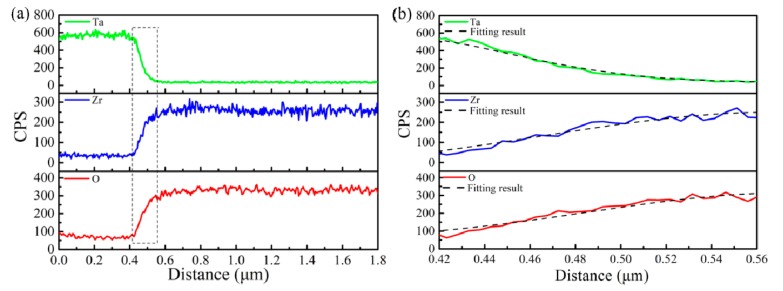
(**a**) Distributions of Ta, Zr and O elements, and (**b**) the fitting results in transition zone analyzed by Fick’s second law.

**Figure 5 materials-13-01265-f005:**
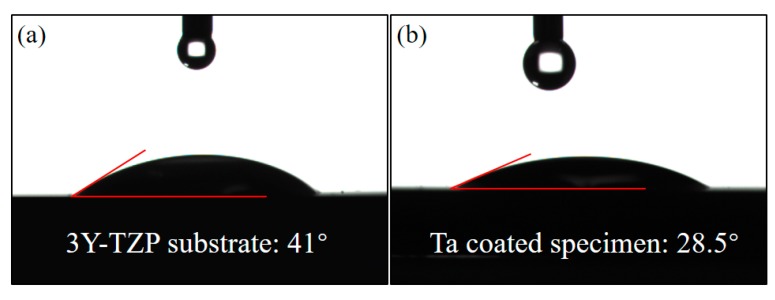
The water contact angle of (**a**) uncoated 3Y-TZP substrate and (**b**) Ta coated specimen.

**Figure 6 materials-13-01265-f006:**
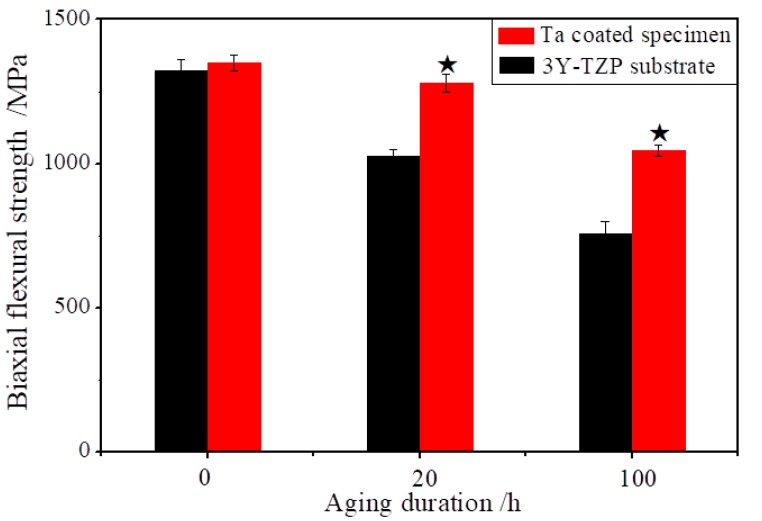
The effect of Ta coatings on the biaxial flexural strength after hydrothermal aging duration 20 h and 100 h, ^★^*p* < 0.05 compared to control.

**Figure 7 materials-13-01265-f007:**
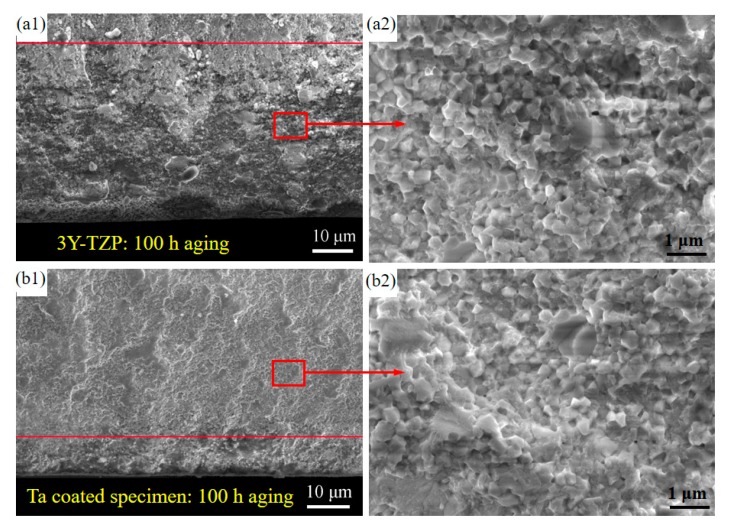
SEM images of fracture surfaces of (**a**1,**a**2) uncoated specimens (control) and (**b**1,**b**2) Ta-coated specimens after 100 h aging duration. Here, the parts below the red line are the *t*→*m* phase transformation region.

**Figure 8 materials-13-01265-f008:**
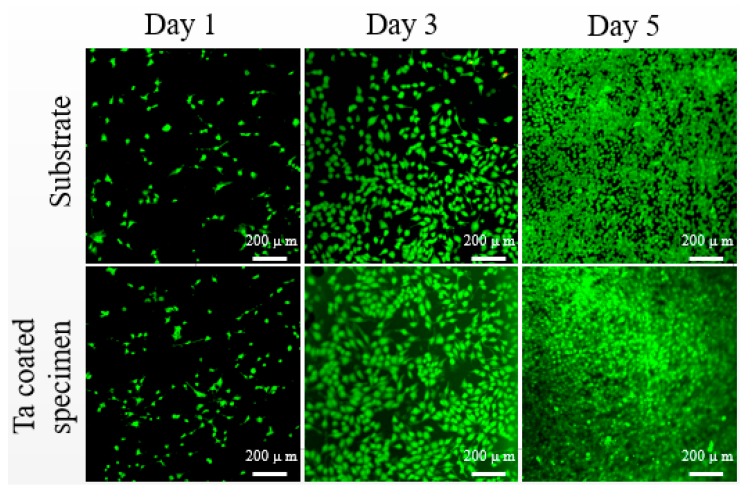
Fluorescence images of Live/Dead staining of osteoblasts culturing on the uncoated and Ta coated specimens for 1, 3 and 5 days.

**Figure 9 materials-13-01265-f009:**
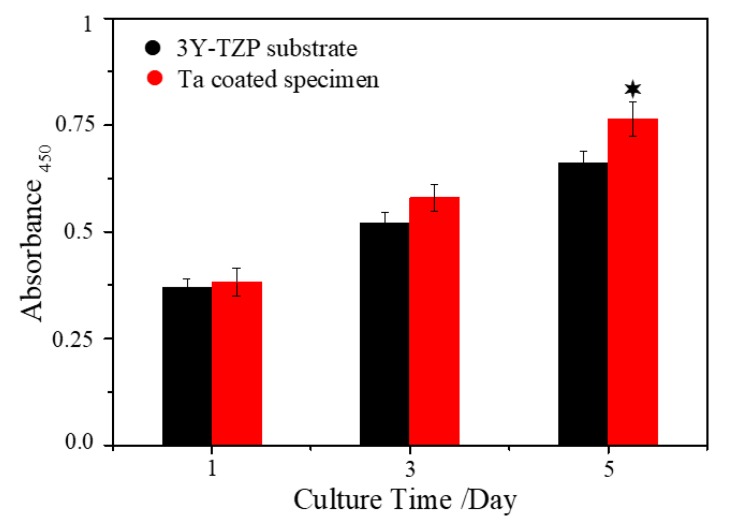
Proliferation results of osteoblasts on the uncoated and Ta coated specimens for 1, 3 and 5 days: ^✶^*p* < 0.05 compared to control.

**Table 1 materials-13-01265-t001:** Preparation parameters of Ta coating.

Process Parameters	Ta Coating
Flow rate of Ar (FAr/sccm)	60
Substrate temperature (ST/°C)	800 ± 5
Deposition pressure (P/Pa)	35
Deposition time (T/min)	10
Cathode voltage of substrate (U/V)	400 ± 10
Cathode voltage of target (U/V)	650 ± 10

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
