# Peer review of "Preparation, Microstructure, Mechanical Properties and Biocompatibility of Ta-Coated 3Y-TZP Ceramic Deposited by a Plasma Surface Alloying Technique"

_materials, 2020, doi:10.3390/ma13061265_

Round 1

Reviewer 1 Report

My comments:

The manuscript requires intensive English proof-reading.

Methodological section is poor:

a) Details on XRD, SEM, EDX measurements are required.

b) Methodology for cell studies has to be improved: brief protocols for live-dead assay (concentrations, timing), LSM and CCK-8 assay (procedure, timing).

c) what was the initial cell seeding density per surface area of the coating? was it always the same? how many experiments have been conducted? "n" numbers have to be shown for each result and graph. 

d) methodology on experimental plan is missing: how many independent experiments have been performed / number of technical replicates, tests for statistical significance.

Results:

a) please provide more detailed explanations in the figure captions.

b) Figure 2 - the peaks for Ta are broaden as compared to zirconium dioxide. What would be the reason? Have you quantified the fraction of amorphous and crystalline phases in the coating?

c) Line 97 "...of 3Y-TZP synthesized at ...".

d) provide explanations to every letter used in the equations also indicating their dimensions.

e) Section 3.2. the loss in strength after 20 h for a Ta-uncoated sample (control) is far higher then 4.1%. Just based on the visual measurement I expect to see around 26%. What is actually a real value of strength loss? How does this accelerated aging correlate to real situation? Does 100 h correspond to 5 years in vivo or you have other values available? "n" value for this result has to be shown in the figure caption indicating statistical significance (if not significant - "NS").

f) Lines 167-168 - this statement is just a speculation. You do not show a proof for this. If you have these result, please show them. If not, remove the statement.

Section 3.3:

a) Please provide quantification of the live-dead images. Otherwise you cannot state that there is no dead cells. Please write for the live-dead images that they are "representative".

b) Line 180 - you write "no significant difference". Please provide statistics showing the p value for these results. Otherwise remove this statement.

c) Fig. 8 and Fig. 9 are mixed in the text (line 182 and 185).

d) Lines 181 and 182. I see only a correlation between the days, but not within the samples.

e) "n" value for the Figure 8 has to be shown.

f) Figure 9. provide the LSM results for a control group (e.g. 3Y-TZP ceramics). Otherwise, this result is not complete.

g) line 201 - to what property? indicate hydrophilic properties of the surfaces, e.g. water contact angle or surface energy.

h) lines 203-205 are just speculations. Please provide more information in this regard.

Author Response

Dear Editor,

Thank you and all reviewers for your comments and the useful suggestions of our manuscript. We have modified the manuscript accordingly, and detailed corrections are listed below point by point:(Please see the attachment).

Author Response

Dear Editor,

Thank you and all reviewers for your comments and the useful suggestions of our manuscript. We have modified the manuscript accordingly, and detailed corrections are listed below point by point:(Please see the attachment)

Reviewer 3 Report

This is a study on a new method to coat Ta on the surface of 3Y-TZP ceramic. The coating was characterized by SEM observation of the cross-section and the surface, XRD, mechanical testing, wettability examination, and the cytocompatibility test using MG-63 cells. The experiments were appropriately designed and conducted. The manuscript was very nicely written. I felt that the manuscript was worth publication after minor revisions.

Minor issues:

Intro and discussion

Well written, but if authors mentioned potential application of the material, it would be great. The property required for the materials depends on how the materials will be used (in clinical practice?).

Line 104

Information of supplier of SPSS should be described.

Line 148

The first author of [23] would be Masahiro Yoshimura. Yoshimura etc. [23] proposed… would be nicer.

Line 177

Again, Yoshimura instead of Masahiro would be better.

Again, I felt that the manuscript was nicely written and there would be only a couple of minor issues to cope with before acceptance.

Additional comments: For cytocompatibility test

Zirconia itself is bioinert materials. We could predict that 3Y-TZP would not bring no extreme results in cytocompatibility test.

To examine the degree of cytocompatibility of Ta coating, not just as a comparison with 3Y-TZP, they need to set appropriate negative control and/or a positive control to evaluate how the Ta coating affect MG63 cells.   For figure 3 It was unclear what the yellow line in Fig3b indicates.   For figure 7 and line 188-200 they discussed the interpretation of SEM images obtained after aging process. But I am not sure if the images and explanation can convince readers that the findings represents inhabitation of t-m phase transformation. Overall, I felt that the proof that directly explain that Ta coat did inhibit t-m transformation was somehow weak.

Author Response

Manuscript Number: materials-679661
Title: Preparation, Microstructure, Mechanical Property and Biocompatibility of Ta-Coated 3Y-TZP Ceramic Deposited by Plasma Surface Alloying Technique
  Dear Editor,
Thank you and all reviewers for your useful suggestions of our manuscript. We have modified the manuscript accordingly, and detailed corrections are listed below point by point: 
Comments and Suggestions for Authors
This is a study on a new method to coat Ta on the surface of 3Y-TZP ceramic. The coating was characterized by SEM observation of the cross-section and the surface, XRD, mechanical testing, wettability examination, and the cytocompatibility test using MG-63 cells. The experiments were appropriately designed and conducted. The manuscript was very nicely written. I felt that the manuscript was worth publication after minor revisions.

Minor issues:
Intro and discussion
Well written, but if authors mentioned potential application of the material, it would be great. The property required for the materials depends on how the materials will be used (in clinical practice?).
 Answer: Actually, the biomedical properties of pure Ta have been reported by many references. So, in this paper, we hope to provide a way to fabricate the pure Ta coating on 3Y-TZP ceramic’s surface. It is the Plasma surface alloying technique (PSAT). Moreover, we also try to make modification according to the reviewer’s suggestion, which is “which has been applied as the implant fixture” in line 36.
Line 104: Information of supplier of SPSS should be described.
 Answer: A good suggestion! We had modified as “The statistical product and service solutions (SPSS Version 19.0, IBM) software”.
Line 148: The first author of [23] would be Masahiro Yoshimura. Yoshimura etc. [23] proposed… would be nicer.
 Answer: A good suggestion! We have utilized Yoshimura to instead Masahiro.
Line 177: Again, Yoshimura instead of Masahiro would be better.
 Answer: We have made modification about it.
Again, I felt that the manuscript was nicely written and there would be only a couple of minor issues to cope with before acceptance.
 Answer: Thank you! All the grammars have been checked, and the manuscript has been polished. The modified manuscript has been provided.
Additional comments: For cytocompatibility test. Zirconia itself is bioinert materials. We could predict that 3Y-TZP would not bring no extreme results in cytocompatibility test. To examine the degree of cytocompatibility of Ta coating, not just as a comparison with 3Y-TZP, they need to set appropriate negative control and/or a positive control to evaluate how the Ta coating affect MG63 cells.  For figure 3 It was unclear what the yellow line in Fig3b indicates.   For figure 7 and line 188-200 they discussed the interpretation of SEM images obtained after aging process. But I am not sure if the images and explanation can convince readers that the findings represents inhabitation of t-m phase transformation. Overall, I felt that the proof that directly explain that Ta coat did inhibit t-m transformation was somehow weak.
 Answer: Thank you! The line composition analysis of Fig. 4 is the yellow line in Fig3b, which was utilized to calculated diffusion of Ta, Zr and O elements. So, we modified it in the manuscript as “and the yellow line in (b) is utilized to line composition analysis in Fig. 4” in line 124-125.
The fracture of SEM images in Fig. 7 is a supplement for the result obtained after aging process in Fig. 6, which is a basic analysis of fracture and also be utilized by many researchers such as in refs. [5], [23] and [27]. Actually, the distinguished effect of Ta-coating of t-m phase transformation had been obviously displayed in Fig. 6.  
Thank you again for these great suggestions!

Reviewer 4 Report

Dear authors, 

I would add some references in the methods of the EDS SEM https://www.ncbi.nlm.nih.gov/pubmed/31060232 

and also I would move the part of the methods of the citotoxicity which are now in the results, in the methods section and in the results and discussion I would also mention the several surface tratments of the implant surfaces such as anatase (https://www.ncbi.nlm.nih.gov/pubmed/29288072) 

  More in detail, I would suggest to the authors to highlight the characteristic of the used method, which is very important in case of medical devices with a peculiar composition.   

In addition, as I have wrote, the part regarding the cytotoxicity, that is currently in the results, should be moved in the method to add more consistency to the manuscript.   

finally other metal surfaces used in medicine and their modification should be mentioned and compared in the discussion. 

Author Response

Manuscript Number: materials-679661

Title: Preparation, Microstructure, Mechanical Property and Biocompatibility of Ta-Coated 3Y-TZP Ceramic Deposited by Plasma Surface Alloying Technique

Dear Editor,

Thank you for these comments and useful suggestions of our manuscript from a biomedical view. We have try to modify the manuscript accordingly, and detailed corrections are listed below point by point:

Comments and Suggestions for Authors

Dear authors, 

I would add some references in the methods of the EDS SEM:

    https://www.ncbi.nlm.nih.gov/pubmed/31060232 

and also I would move the part of the methods of the citotoxicity which are now in the results, in the methods section and in the results and discussion I would also mention the several surface treatments of the implant surfaces such as anatase:

https://www.ncbi.nlm.nih.gov/pubmed/29288072

Abstract: Implantology research framed the implant surface as a key element for a good and sustainable osseointegration of an implant fixture. The aim of this study was to analyze the antibacterial properties of anatase-coated titanium healing screws through microbiological and scanning electron microscopy. The comparison of the bacterial colonies growth between the anatase-coated titanium healing screws and non-coated titanium healing screws showed comparable antibacterial properties, without significant statistical differences. The scanning electron microscopy observations confirmed the microbiological study. These data, also considering previous reports on the positive effects on osteoblasts genetic expressions, might suggest a use of the anatase-coated titanium healing screws to preserve the tissues surrounding implants from microbial attacks.

Answer: A great suggestion! We had modified and cited these two papers as: ref. 9 and 10.

[9] Bernardi, S.; Bianchi, S.; Tomei, A.R.; Continenza, M.A.; Macchiarelli, G., Microbiological and SEM-EDS Evaluation of Titanium Surfaces Exposed to Periodontal Gel: In Vitro Study. Materials (Basel) 2019, 12(9): 1448-1460.

[10] Bernardi, S.; Bianchi, S.; Botticelli, G.; Rastelli, E.; Tomei, A.R.; Palmerini, M.G.; Continenza, M.A.; Macchiarelli, G., Scanning electron microscopy and microbiological approaches for the evaluation of salivary microorganisms behaviour on anatase titanium surfaces: In vitro study. Morphologie 2017, 102(336):1-6.

More in detail, I would suggest to the authors to highlight the characteristic of the used method, which is very important in case of medical devices with a peculiar composition. In addition, as I have wrote, the part regarding the cytotoxicity that is currently in the results should be moved in the method to add more consistency to the manuscript. 

Answer: Thank you for the suggestions! We hope to express as the original pattern.

Finally other metal surfaces used in medicine and their modification should be mentioned and compared in the discussion. 

Answer: Actually, the biomedical properties of pure Ta have been reported by many references. So, in this paper, we hope to provide a way to fabricate the pure Ta coating on 3Y-TZP ceramic’s surface by traditional process. It is the Plasma surface alloying technique (PSAT).

Thank you again for these great suggestions!

Reviewer 5 Report

In this manuscript, the development of more hydrophilic biomedical implants  favorises their cytocompatibility.Tantalum is considered a biocompatible metal because a thin layer of oxide passives the metal surface, and a recent paper (Brüggemann et al, J.Bone Joint Surg Am, 2020) indicates the presence of low level of tantalum in blood of orthopaedic patients. However, in 2014 (Babis et al, Acta Orthopaedica, 85:667) an important case of metallosis was observed because in the presence of mechanical abrasion, the oxide coating can be removed and the capacity of the metal to be repaired is lower in the absence of free oxygen, as in the case of orthopaedic implants. A commentary on this subject would be useful

Author Response

Manuscript Number: materials-679661
Title: Preparation, Microstructure, Mechanical Property and Biocompatibility of Ta-Coated 3Y-TZP Ceramic Deposited by Plasma Surface Alloying Technique
Dear Editor,
Thank you for the useful comments and suggestions. We try to modify the manuscript accordingly, and detailed corrections are listed below point by point: 
Comments and Suggestions for Authors
In this manuscript, the development of more hydrophilic biomedical implants favorites their cytocompatibility. Tantalum is considered a biocompatible metal because a thin layer of oxide passives the metal surface, and a recent paper (Brüggemann et al, J.Bone Joint Surg Am, 2020) indicates the presence of low level of tantalum in blood of orthopaedic patients. However, in 2014 (Babis et al, Acta Orthopaedica, 85:667) an important case of metallosis was observed because in the presence of mechanical abrasion, the oxide coating can be removed and the capacity of the metal to be repaired is lower in the absence of free oxygen, as in the case of orthopaedic implants. A commentary on this subject would be useful.
Answer: A great suggestions! Your comments are appreciated by all authors. Actually, the biomedical properties of pure Ta have been reported by many references. So, in this paper, we hope to provide a way to fabricate the pure Ta coating on 3Y-TZP ceramic’s surface, because Ta coating is hardly to fabricate on 3Y-TZP by traditional process. That is a metallization of 3Y-TZP Ceramic surface by pure Ta via Plasma surface alloying technique (PSAT). Moreover, although the recent paper (Brüggemann et al, J.Bone Joint Surg Am, 2020) was not found due to the problem of the Journal’s official website system, another paper of [Babis et al, Acta Orthopaedica, 85:667] had been cited in manuscript as ref. [14].
Thank you again for these great suggestions!  

Round 2

Reviewer 1 Report

General: The authors improved the manuscript based on my comments. However, not all were considered:

Even in the revised version there are many mistakes. The authors have to ask a native speaker for English corrections. The authors deleted the Figure on cytoskeleton using LSM. Their excuse is that they do not have sufficient time to do additional experiments because of Chinese New Year. This is not actually an excuse, since I had to revise his work during holidays! I have to see the results anyway. I asked for quantification of live-dead images shown on Figure 7. Again, the excuse deals with Chinese New Year. But with this point I will be ok, if the authors write in the figure caption "Representative fluorescence images of ....".

Author Response

Dear reviewer,

On behalf of my co-authors, we thank you very much for giving an opportunity to revise our manuscript, we appreciate editor for positive and constructive comments and suggestions on our manuscript entitled “Effect of Ta coating on the mechanical property and biocompatibility of 3Y-TZP ceramic”. Those comments are all valuable and very helpful for revising and improving our paper, as well as the important guiding significance to our researches. According to the aim of this manuscript, we changed the title to” Preparation, Microstructure, Mechanical Property and Biocompatibility of Ta Coating on 3Y-TZP Ceramic Deposited by Plasma Alloying Technique”.

Actually, it is not easily to the preparation of Ta coating on the 3Y-TZP ceramic. The fabrication of Ta coating using plasma alloying technique plays a key role in this article. Moreover, the microstructure and diffusion were investigated in this manuscript. The Ta element had not reactive with the 3Y-TZP ceramic but diffusion. The effect of Ta coating on the mechanical property was studied. Due to the effect of inhabitation the tm phase transformation of 3Y-TZP by Ta coating, the biaxial flexural strength caused by the phase transformation during hydrothermal aging is reduced e.g. p < 0.05 after 20 h and/or 100 h. In addition, the biocompatibility of Ta-coated specimens was also investigated. Actually, the biocompatibility and performance of Ta coating is the same as pure Ta, which can be found in many previous reports. So, we just carried out several experiments to show Ta-coating’s biocompatibility.

We tried our best to revise our manuscript according to the comments. Revised portion are marked in the paper.

Special thanks to you for your good comments.

Your sincerely,

Y.S. Wang

Institute of New Carbon Materials,

Taiyuan University of Technology, Taiyuan 030024, P. R. China

Reviewer 2 Report

Review of “Effect of Ta Coating on the Mechanical Property and Biocompatibility of 3Y-TZP Ceramic” (revision 1), by Ke Zheng et al.

General notes: 

The authors have greatly improved the manuscript.  This reviewer appreciates the addition of contact angle measurements (the new Figure 5), and strongly encourage the authors to perform statistical analyses on these data as well.

However, two small edits must be done before it can be accepted (see Questions 1 and 4).  In addition, some of the typos may be corrected during the copyediting process, but the authors may want to hire someone to rewrite parts of the paper so it can be read in English more easily.

Responses to Questions:

Question 1:

The reviewer thanks the authors for correcting the manuscript as requested.  However, please delete “/or”.  It should read “… after 20 h and 100 h.”)

Question 2:

The reviewer thanks the authors for correcting the manuscript as requested. 

Question 3:

The reviewer thanks the authors for correcting the manuscript as requested. 

Question 4:

Thank you for clarifying the figure.  Please be sure to update the caption of Figure 3(b)-(f) to note that the Ta coating shown is 390 nm thick.

Question 5:

The reviewer thanks the authors for correcting the manuscript as requested. 

Question 6:

The reviewer thanks the authors for correcting the manuscript as requested. 

Question 7:

The reviewer thanks the authors for expanding the manuscript as requested. 

Question 8:

The reviewer thanks the authors for providing p-values as requested.

Question 9:

The reviewer thanks the authors for expanding the manuscript as requested. 

Author Response

Dear reviewers and editor,

On behalf of my co-authors, we thank you very much for giving an opportunity to revise our manuscript, we appreciate editor for positive and constructive comments and suggestions on our manuscript entitled “Effect of Ta coating on the mechanical property and biocompatibility of 3Y-TZP ceramic”. Those comments are all valuable and very helpful for revising and improving our paper, as well as the important guiding significance to our researches. According to the aim of this manuscript, we changed the title to” Preparation, Microstructure, Mechanical Property and Biocompatibility of Ta Coating on 3Y-TZP Ceramic Deposited by Plasma Alloying Technique”.

Actually, it is not easily to the preparation of Ta coating on the 3Y-TZP ceramic. The fabrication of Ta coating using plasma alloying technique plays a key role in this article. Moreover, the microstructure and diffusion were investigated in this manuscript. The Ta element had not reactive with the 3Y-TZP ceramic but diffusion. The effect of Ta coating on the mechanical property was studied. Due to the effect of inhabitation the tm phase transformation of 3Y-TZP by Ta coating, the biaxial flexural strength caused by the phase transformation during hydrothermal aging is reduced e.g. p < 0.05 after 20 h and/or 100 h. In addition, the biocompatibility of Ta-coated specimens was also investigated. Actually, the biocompatibility and performance of Ta coating is the same as pure Ta, which can be found in many previous reports. So, we just carried out several experiments to show Ta-coating’s biocompatibility.

We tried our best to revise our manuscript according to the comments. Revised portion are marked in the paper.

Special thanks to you for your good comments.

Your sincerely,

Y.S. Wang

Institute of New Carbon Materials,

Taiyuan University of Technology, Taiyuan 030024, P. R. China
